# *Chlorella vulgaris* and *Tetradesmus obliquus* Protect Spinach (*Spinacia oleracea* L.) against *Fusarium oxysporum*

**DOI:** 10.3390/plants13121697

**Published:** 2024-06-19

**Authors:** Catarina Viana, Méanne Genevace, Florinda Gama, Luísa Coelho, Hugo Pereira, João Varela, Mário Reis

**Affiliations:** 1Faculty of Sciences and Technology, University of Algarve, Campus of Gambelas, 8005-139 Faro, Portugal; jvarela@ualg.pt (J.V.); mreis@ualg.pt (M.R.); 2GreenCoLab-Associação Oceano Verde, University of Algarve, Campus of Gambelas, 8005-139 Faro, Portugal; florindagama@greencolab.com (F.G.); lcoelho6@gmail.com (L.C.); hugopereira@greencolab.com (H.P.); 3Faculty of Environmental Innovations, HAS University of Applied Sciences, Onderwijsboulevard 22, 15223 DE ’s-Hertogenbosch, The Netherlands; 4Centre of Marine Sciences, Faculty of Sciences and Technology, University of Algarve, Campus of Gambelas, 8005-139 Faro, Portugal; 5MED—Mediterranean Institute for Agriculture, Environment and Development, University of Algarve, Campus of Gambelas, 8005-139 Faro, Portugal; 6CHANGE—Global Change and Sustainability Institute, Instituto de Investigação e Formação Avançada, Universidade de Évora, Pólo da Mitra, Ap. 94, 7006-554 Évora, Portugal; 7CHANGE—Global Change and Sustainability Institute, Faculdade de Ciências e Tecnologia, Universidade do Algarve, Campus de Gambelas, 8005-139 Faro, Portugal

**Keywords:** biocontrol, phytopathogenic fungi, microalgae, soilborne disease, in vivo, in vitro

## Abstract

*Chlorella vulgaris* and *Tetradesmus obliquus* were tested as biocontrol agents against the phytopathogenic fungus *Fusarium oxysporum*. This evaluation was conducted through in vitro and in vivo trials with spinach (*Spinacia oleracea* L.). The in vitro trials showed that *C. vulgaris* and *T. obliquus* were able to inhibit the phytopathogen, showing a similar inhibitory effect to that of the positive controls (Rovral, BASF^®^ and Biocontrol T34, Biocontrol Technologies^®^ S.L.). *C. vulgaris* aqueous suspensions at 3.0 g L^−1^ led to a hyphal growth of 0.55 cm, each corresponding to a reduction of 63% of fungal growth. With *T. obliquus*, the hyphal growth was 0.53 cm when applied at a concentration of 0.75 g L^−1^, having an inhibition of fungus growth of 64%. Thereafter, these results were validated in an in vivo trial on spinach using the same controls. The results revealed a lower severity and disease incidence and a reduction in the disease’s AUDPC (area under the disease progress curve) when spinach was treated with the microalgae suspensions. Overall, these findings highlight the potential of *C. vulgaris* and *T. obliquus* suspensions as promising biocontrol agents against *F. oxysporum* in spinach when applied through irrigation.

## 1. Introduction

In 2050, the world population is estimated to have grown to nine billion, increasing food demand by 70% [1]. If pests and plant diseases are not managed, the global yield loss in the major crops could reach up to 50%. When no crop protection tools are used, this could eventually add up to 75% [2]. Crop protection is primarily based on synthetic chemical products. The use of these chemicals has led to increased and stabilized yields as well as improved crop quality [3]. However, their excessive use has led to the emergence of resistance and the loss of effectiveness of agrochemicals against phytopathogens [3,4]. In addition, this misuse has caused significant impacts on human health and the environment, leading to water pollution, loss of biodiversity, and soil degradation [5]. The European Green Deal has ongoing initiatives like the Farm to Fork and Biodiversity strategies, where the target by 2030 is to reduce the usage of chemical pesticides and fertilizers by 50% and 20%, respectively [6]. Ultimately, the goal is to reduce the impact of pesticides and fertilizers on the environment and human health [7] while maintaining production levels. Therefore, research on solutions for this problem has been focused on improving integrated pest management techniques together with the development of alternative pest and disease control [7].

Biopesticides can aid in accomplishing this task by implementing an environmentally friendly approach. Their multiple modes of action reduce the acquisition of resistance by plant pathogens [8]. Additionally, biopesticides are applied in lower concentrations and are biodegradable [8,9]. There are several different sources of biopesticides, such as beneficial fungus (e.g., *Trichoderma* spp.), which are non-pathogenic soilborne fungi that are able to colonize plant roots, forming a symbiotic relationship with the plant. In addition, they are also known for their activity in the control of phytopathogenic fungi, such as *Sclerotium rolfsii*, *Rhizoctonia solani*, and *Fusarium oxysporum* [10,11,12]. The effect of this fungi as a biopesticide has been reported in in vitro and in vivo trials [11,13].

Previous studies have shown the potential of algae-based products as biostimulants and as biocontrol agents for agriculture [14] due to their several bioactive compounds, such as amino acids, polypeptides, antioxidants, enzymes, phytohormones, vitamins, phenolic compounds, allelopathic chemicals, and carotenoids [8,15,16,17,18,19,20]. These compounds might enhance plant resistance by direct antagonism or via a positive effect on beneficial soil microorganisms that stimulate nutrient absorption and, thus, improve plant growth [21,22]. Therefore, algae have been indicated as viable candidates for biocontrol. 

In vitro biocontrol activities of *Chlorella protothecoides*, *Chlorella vulgaris,* and *Tetradesmus obliquus* (microalgae) suspensions, grown in piggery wastewater, against *F. oxysporum* have already been evaluated. *C. vulgaris* was shown to exert an in vitro inhibition higher than 40% for all the concentrations tested [23]. In another study, *C. vulgaris* grown in wastewater was able to decrease the growth of fungi such as *R. solani* and *F. oxysporum*, as well as of oomycetes, i.e., *Phytophthora capsica* and *Pythium ultimum* [24]. Water-based extracts of *T. obliquus* have also displayed in vitro antifungal activities against *Sclerotium rolfsii*, resulting in growth inhibition of up to 32.0% [14]. Scaglioni et al. [25] reported that carotenoid and phenolic extracts of *Nannochloropsis* sp. and *Spirulina* sp. inhibit mycelium growth of *F. graminearum*. To the best of our knowledge, however, no published studies of in vivo trials were published to validate these previous in vitro studies.

*Fusarium oxysporum* Schlechtendal is a complex species of soilborne fungi that can be found worldwide [26]. This species affects a large range of hosts, including plant pathogens and human pathogens [26]. As a phytopathogenic fungus, it is responsible for fusarium wilt, which leads to seedling damping off and wilt and rot disease in mature plants, causing browning of the vascular tissues, yellowing of the leaves and even plant death [27,28]. Chlamydospores from *Fusarium oxysporum* can persist in dormancy and infect the soil for many years [28]. This species is among the most harmful fungal plant pathogens, affecting a wide range of crops (cotton, tomatoes, strawberries, lettuce, and bananas), as well as ornamental crops (orchids and gerbera) and even weeds or parasitic plants (broomrape and witchweed) [26,29,30]. The ability to infect several species of plants might be due to its gene-for-gene relationships allowing many specific forms for each host, being called formae speciales [26,30,31,32,33]. More than 150 formae speciales have been identified so far [30].

Spinach (*Spinacia oleracea* L.) is an economically important vegetable with a high-value composition rich in several nutrients and carotenoids and is a source of vitamin C, calcium, and iron [34]. In addition, spinach has a rich composition in phenolic compounds and flavonoids, having at least 13 different types of flavonoids that provide an antioxidant and anticancer effect [35]. According to Gorelick et al. [36], spinach contains several components that directly influence human health, such as ecdysteroids and hormones known for their anti-inflammatory properties. Spinach is also susceptible to *F. oxysporum* and is often used as a model species in phytopathogenic in vivo trials [37]. Due to its high functional and genetic diversity, the strain responsible for fusarium wilt in spinach belongs to forma specialis *spinaciae* W.C. Snyder & H. N. Hansen [26,28,33,38].

The increasing resistance of pathogens to many synthetic pesticides and the growing demand from governments and the public for the more sustainable control of pests and diseases has raised the following question: will it be possible to use microalgae as biocontrol agents against *F. oxysporum*?

In this context, the main objective of this work was to evaluate the biocontrol potential of *C. vulgaris* and *T. obliquus* against *F. oxysporum*. The optimal concentrations for each microalga were previously tested in vitro and afterwards validated in in vivo assays on spinach.

## 2. Results and Discussion

### 2.1. In Vitro Trials

To ensure that the suppressive effect was related to the algae suspensions under study and not to the substrate (peat) microbiome, its microbiota with and without a thermal treatment was analysed (Table 1).

Table 1 presents the results obtained for the microbiological evaluation of the substrates before use. As expected, microorganisms were detected only on the non-treated substrate. Therefore, the results obtained for the thermally treated substrate were expected not to have been influenced by the microbiota in the substrate.

Figure 1 shows the results obtained in the in vitro assays regarding the biocontrol potential of *C. vulgaris* and *T. obliquus* against *F. oxysporum*. Plotting the concentrations of *C. vulgaris* and concentrations of *T. obliquus* ratios versus *F. oxysporum* hyphal growth led to two quadratic models.

With *C. vulgaris*, the increase in concentration led to a decrease in the growth of *F. oxysporum* hyphae. The concentrations of 2.0, 2.5, and 3.0 g L^−1^, showed the lowest growth of 0.63, 0.57, and 0.55 cm, respectively. In the negative control (water), the growth of *F. oxysporum* was 1.49 cm, leading to a decrease of 0.94 cm when treated with 3.0 g L^−1^ of *C. vulgaris*. At this concentration (3.0 g L^−1^), a hyphae growth inhibition of 57.3% was calculated. The positive controls Rovral and T34, tested against *F. oxysporum*, displayed a hyphal growth of 0.18 ± 0.05 cm and 0.58 ± 0.06 cm, which correspond to growth inhibitions of 86 ± 11 % and 53 ± 10%, respectively. In addition, apart from the concentration of 0.01 g L^−1^, all treatments induced more than 20% pathogen hyphae growth inhibition, and the highest inhibitions occurred in the treatments where microalgae biomass was applied at a higher concentration.

Similar findings were found in a study [22] on the potential of *C. vulgaris* at concentrations varying between 0.6 and 5.0 g L^−1^ against *F. oxysporum*, where inhibition ranged between 40 and 75%. In this case, the authors used different organic solvents that are known to have a higher capacity to extract bioactive compounds compared to only water-based extracts. Additionally, Vehapi et al. [39] reported the inhibition of *F. oxysporum* growth when treated with an oil extract of *C. vulgaris* (0.25, 0.5, 0.75, and 1.0 g L^−1^) cultivated in three different media: Iroko tree extract water medium; bald basal medium; and Istanbul Water and Sewage Administration medium.

Finally, when treated with 1.0, 2.5, and 5.0 g L^−1^ of *C. vulgaris* obtained from piggery wastewater, there was an inhibition by up to 40% of *F. oxysporum* growth [23].

When treated with *T. obliquus*, the lower concentrations presented a higher effect on hyphal growth (Figure 1). With this alga, when applied at a concentration of 0.75 g L^−1^, the fungus only grew 0.53 cm, which corresponds to an inhibition of 64%. These were better results than those observed by Ferreira et al. [23], where the application of *T. obliquus* led to inhibitions of 46.8% when applied at the concentration of 5.0 g L^−1^. On the other hand, in the same figure, Figure 1, it should be noted that higher concentrations stimulated the growth of the fungus by up to 12% when 2.5 g L^−1^ of *T. obliquus* was applied. This might be explained by the increase in nutrient content that the fungus can use. Similar results were obtained in the study by Schmid et al. [14], where treatment with these microalgae at a concentration of 2.0 g L^−1^ led to an increased growth of *Alternaria alternata*.

Although there are already several studies on the biocontrol effect of algae against phytopathogens, the mechanisms behind this are not yet understood. According to Scalioni et al. [25], this effect may be related to bioactive compounds with a phenolic acid profile that have various antimicrobial properties, such as chlorogenic acid. Other studies suggest that this antimicrobial activity is due to carotenoid pigments [40,41]. However, in this study, aqueous suspensions of the dry biomass of microalgae induced inhibition over 40% to 50%. In this case, the effect should be related to water-soluble compounds, such as flavonoids, which have already been identified for their various biological activities [14,42].

The obtained results are promising, as there were no statistical differences when compared to T34, the biological positive control. Since both *C. vulgaris* and *T. obliquus* showed promising potential to inhibit *F. oxysporum* growth, an in vivo trial was conducted to validate these results. The concentrations 2.0, 2.5, and 3.0 g L^−1^ of *C. vulgaris* and 0.5, 0.75, and 1.0 g L^−1^ of *T. obliquus* suspensions were selected to be validated in vivo, considering their inhibition percentage, over 40% and 50%, respectively, obtained in vitro.

### 2.2. In Vivo Trials

#### 2.2.1. Effect of *C. vulgaris* Suspensions on *F. oxysporum* f. sp. *spinaciae*

The incidence of the disease was evaluated weekly alongside its severity. Table 2 shows the incidence percentage of the disease by the end of the trial, i.e., 35 days after inoculation (DAI). The treatments with *C. vulgaris* and the negative control (water) used in the in vivo trial did not show a significant difference regarding the incidence of the disease. Therefore, all the treatments showed visual symptoms of the disease caused by *F. oxysporum*, including the biocontrol agent T34 and the commercial synthetic fungicide Rovral.

Interestingly, even when infected, the spinach plants did not show any increase in disease severity when treated with the microalgal suspensions (Figure 2). Quite the opposite, on the non-treated peat (Figure 2A), the synthetic pesticide Rovral and the negative control (water) led to a significantly higher disease severity than other treatments. No significant differences were found between the concentrations of 2.0, 2.5, and 3.0 g L^−1^ of *C. vulgaris* and T34. However, these three concentrations significantly decreased the severity of the disease in spinach.

Upon conducting the same trial with thermally treated peat, *C. vulgaris* induced a significantly lower disease severity when compared to plants treated with Rovral, T34, and water (Figure 2B), except for *C. vulgaris* at 3.0 g L^−1^. No significant differences were found between the concentrations of 2.0, 2.5, and 3.0 g L^−1^ of *C. vulgaris*. These results suggest that *C. vulgaris* significantly decreased disease severity, supporting the hypothesis that only microalgal biomass displayed a suppressive effect on the growth of the fungus in vivo.

Regarding the AUDPC, in both the thermally treated and non-treated peat trials, the plants with Rovral and water treatments had a significantly higher AUDPC than any treatment with algal suspensions (Figure 3), except for the highest concentration used, i.e., 3.0 g L^−1^. Thus, when the alga treatments were used, disease progression in the spinach plants decreased significantly. Using non-treated (and heat-treated) peat as a substrate (Figure 3A), the effect of 2.5 g L^−1^ of *C. vulgaris* on the AUDPC was lower than that of the negative control, suggesting that the treatment with *C. vulgaris* reduced the growth of *F. oxysporum* in spinach. Compared with T34, the treatments of *C. vulgaris* (2.0, 2.5, and 3.0 g L^−1^) showed no significant differences.

With the thermally treated substrate (Figure 3B), the treatments Rovral, T34, and water had a significantly higher AUDPC than when *C. vulgaris* was applied. These algal suspensions, however, yielded no significant differences among each other, indicating that the disease was less aggressive and showed reduced growth when algal suspensions were applied. Once again, as observed for disease severity, the results confirmed that in thermally treated substrates, *C. vulgaris* has the potential to control the growth of *F. oxysporum*. In this case, even though the disease was present in all the treatments, the severity was significantly lower when the spinach was treated with algal suspensions. *C. vulgaris* antifungal properties have been associated with the production of metabolites such as antibiotics, enzymes, proteins, and fatty acids. These compounds have antimicrobial effects, as observed by Perveen et al. [22] and Hamed et al. [43]. The inhibition of the fungus might also be due to the presence of polyphenols during the oxidation of vital compounds [44]. Indeed, the total polyphenolic content of *C. vulgaris* is 73.01 ± 2.6 mg GAE/g DW, which can be one of the possible explanations for the effect of this alga on fungal growth and the aggressivity of the disease on the plants. However, Vehapi et al. [44] have also suggested that the antifungal effect of *C. vulgaris* on *Aspergillus niger*, *Alternaria alternata*, and *Penicillium expansum* might be due to its high contents of terpenes, alkaloids, and polypeptides.

#### 2.2.2. Effect of *T. obliquus* Suspensions on *F. oxysporum*

As for *C. vulgaris*, *F. oxysporum*-induced disease incidence in spinach treated with *T. obliquus* was evaluated on plants grown on two types of substrates (non-treated peat and thermally treated peat). This evaluation is presented in Table 3. When treated with *T. obliquus*, the observed disease incidence after 47 DAI in spinach decreased (Table 3). In non-treated peat, *T. obliquus* reduced the incidence of the disease by 58% when applied at a concentration of 0.5 g L^−1^ with statistical differences to the biocontrol T34 and the fungicide Rovral.

Again, as for the thermally treated peat, the algal suspension reduced disease incidence. With the concentration of 1.0 g L^−1^, the suspension reduced the incidence of the disease and was able to control it.

In Table 3, the difference between the treated and non-treated peat is apparent. The non-treated peat had a lower incidence of the disease when compared with the thermally treated peat. According to Hao and Ashley [45], soil health is a determining factor in the management of soilborne pathogens. Soil communities can induce plant resistance or compete directly with plant pathogens, having an important role in the soil and plant processes and functions [46]. Therefore, the results obtained in the non-treated peat demonstrate that the community present in this substrate, by having more nutrients at its disposal due to the application of *T. obliquus*, led to *F. oxysporum* not being as devastating as in the water treatment (negative control), where this fungus affected all plants. It should be noted that the incidence of the disease was also higher when the synthetic fungicide was applied when compared with the alga treatment, implying that it may have negatively affected the substrate microbiome by weakening the microbiota, giving more space to *F. oxysporum* to develop, which is proven later in this study. However, more studies need to be conducted to confirm the effect of algae on the soil microbiome and its effects on phytopathogenic fungi.

In both substrates, at the end of the assay (47 DAI), there was a significant reduction in disease severity when *T. obliquus* was applied (except for the concentration of 0.75 g L^−1^ in the thermally treated peat) compared with the water treatment (Figure 4). With the addition of the microalgal suspensions, the severity was always lower than level 2 (indicating only minor damage on plants), except for treatment with 0.75 g L^−1^ when the substrate was the thermally treated peat. In this case, the observed disease severity was 2.5.

Once more, it may be observed that the severity of the symptoms caused by the plant pathogen is more in the thermally treated peat. These results can support the idea that a healthy microbiome makes it harder for the fungus to develop. Hao and Ashley [45] also mention that as much as the use of antimicrobial compounds is encouraged, the organic input is even more important because it is positively correlated with the level of disease suppression. By adding the aqueous algae suspensions, we are introducing several compounds that will enrich the substrate, due to the composition of the algae, as already mentioned. In agriculture, it is common to add green manures or organic amendments to improve soil quality. Green algae are great candidates to be good organic amendments due to their photosynthetic capacity since they capture atmospheric CO_2_ and convert it into biomass. Its introduction into the substrate will not only increase the amount of CO_2_ in it but will also increase its microflora and fauna [47]. Some studies suggest that the application of organic materials enhances the natural soil suppressiveness, in which “the phytopathogen does not establish or persist”, reducing the damages ensued from the disease caused by the fungus [48,49].

In the non-treated peat, 47 DAI, the AUDPC in the water treatment was significantly higher than in the algal treatments (Figure 5A). The lowest AUDPC was observed at the concentration of 0.5 g L^−1^, which was significantly lower than all other treatments (Figure 5A).

*Tetradesmus obliquus* at 1.0 g L^−1^ in the thermally treated peat (Figure 5B) had a significantly lower development of the disease than the negative control.

Finally, looking at the positive control Rovral, in the non-treated peat, it turns out that the disease was able to develop throughout the trial, which led to the death of the plants. This indicates that agrochemicals can affect not only the disease, as shown in the graph of the treated substrate (Figure 5B), but can also affect the microbiome and consequently allow the pathogenic fungus to develop without having competition.

As phenolic and polyphenolic compounds have been described in the literature as an alternative to chemical products, *T. obliquus* was analysed for its total polyphenolic content, which reached 53.76 ± 6.4 mg GAE/g DW. This relatively high content in polyphenols might explain how suspensions of this microalga are able to affect fungal growth.

Some studies show that the phenolic compounds and flavonoids extracted from *Barkleyanthus salicifolius* affected the growth of *F. oxysporum* and *Colletotrichum gloeosporioides* by 66.1% and 92.8%, respectively [50]. Algae biomass has a diverse composition, and previous studies suggest that their capacity to inhibit fungal growth is due to their composition of bioactive compounds such as phenolics and tocopherols, among others [8,15]. On the other hand, it is also known that pesticides can act in several ways, such as by disrupting the cytoplasmic membrane, inhibiting enzyme activity, or activating the plant immune system [21]. A high polysaccharide content is also mentioned to be one way of activating the plant immune system [51]. The authors reported that when applied to tomato plants, the effect of crude polysaccharides from *C. vulgaris* increased the expression of pathogenesis-related genes and of those coding for antioxidant enzymes such as *β*-1,3-glucanase, ascorbate peroxidase (APX), and persulfide dioxygenase (PDO). Glucanase, a lytic enzyme, plays an important role in biological control [52,53]. The enzyme exo-α-1,3-glucanase is capable of binding to the cell walls of several phytopathogenic fungi such as *Aspergillus niger*, *Botrytis cinera*, *Coletotrichum acutatum*, *Fusarium oxysporium*, *Penicillium aurantiogriseum*, or *Rhizoctonia solani* [54].

Additionally, microalgae polysaccharides may also increase plant growth and tolerance for abiotic stress, thus acting as biostimulants [55,56].

## 3. Material and Methods

### 3.1. Phytopathogenic Isolates

The trials were conducted at the GreenCoLab laboratories and in a greenhouse of FCT at the Gambelas Campus of the University of Algarve (37°02′35.45′′ N, 7°58′20.64′′ W).

The isolate of the target pathogen used in this experiment (*F. oxysporum* Schlechtendal f. sp. *spinaciae* W.C. Snyder & H.N. Hansen) was supplied from the mycological collection of the Mediterranean Institute for Agriculture, Environmental and Development (MED), collection id KAU9VKKA01N, from the University of Algarve. The fungi were grown on potato dextrose agar media (PDA, Biokar, Allonne, France) for 7 days at 25 ± 2 °C to obtain mycelial discs as inoculum.

### 3.2. Microalgal Biomass Cultivation

The dry biomass of *Chlorella vulgaris* and *Tetradesmus obliquus* was obtained from Allmicroalgae Natural Products S.A. (Leiria, Portugal). Cultures were grown in tubular photobioreactors using the company’s standard industrial protocols. The biomass was harvested by membrane filtration, spray-dried into a fine powder, and stored in bags containing an oxygen absorber.

### 3.3. In Vitro Trial to Determine Inhibition Percentage of Microalgae against F. oxysporum

The peat microbiome was determined through the enumeration of fungi and bacteria in adequate culture media. Samples were suspended in phosphate-buffered saline (PBS), and serial decimal dilutions were prepared and then inoculated into a culture media suitable for the tested microorganism growth. For fungi, aerobic bacteria, and actinomycetes enumeration, culture media were inoculated by the spread-plate technique. Fungi were cultivated on potato dextrose agar (PDA) (Biokar, France) plates and incubated at 25 ± 2 °C and 55 ± 2 °C for 24–48 h; total aerobic bacteria on plate count agar (PCA) (Oxoid, England) at 25 ± 2 °C and 55 ± 2 °C for 24–48 h; and actinomycetes on 1/2PCA at 25 ± 2 °C and 55 ± 2 °C for 24–48 h [57]. These assays were carried out in triplicate.

The method used is an adaptation of the diffusion method from Ambika and Sujatha [58] and Machado et al. [59]. Potato dextrose agar (PDA) was used as the growth medium (39 g L^−1^) and supplemented with the antibiotic chloramphenicol (0.048 g L^−1^) to minimize other microorganisms that might otherwise grow on the medium. The medium was sterilized and distributed into 90 mm Petri dishes in a UV sterile flow chamber (FASTER, BH-EN 2005). Algae suspensions with different concentrations (0.01, 0.1, 0.25, 0.5, 0.75, 1.0, 1.25, 1.5, 2.0, 2.5, 3.0, and 5.0 g L^−1^) of *C. vulgaris* and *T. obliquus* were made using sterile deionized water. As positive controls, Rovral (iprodione, BASF^®^, Ludwigshafen, Germany) and T34 (*Trichoderma asperellum,* Biocontrol Technologies^®^, Spain) were used as recommended by the manufacturers at the concentrations of 1.0 and 0.01 g L^−1^, respectively. Sterile deionized water was used as the negative control. For each treatment, 450 µL of the algal suspension was placed on separate PDA plates and spread evenly using a cell spreader. All procedures were performed under aseptic conditions. During the in vitro trials, fifteen treatments were used for both algae: twelve algal concentrations and three controls. Each treatment consisted of three or five replicates. The inoculation of the PDA with the phytopathogenic fungi was obtained with a seven-day-old *Fusarium oxysporum* pure culture, from which 6.5 mm discs were prepared. Each disc was placed in the centre of each Petri plate. After the inoculation, the PDA plates were stored in the incubation chamber at 25 ± 2 °C for 72 h, being removed every 24 h to measure the radius of fungal mycelium growth. After 72 h, the inhibition percentage of the fungal growth was analysed and compared to that of the negative control (water). The inhibition percentage of the fungal growth was also calculated according to the following formula:$$\mathrm{IP}\;(\%)=\frac{\left(Rc-R1\right)}{Rc}\times 100,$$
where *R_c_* = the radius of the growth zone (cm) of the pathogen that grew on a plate containing only water and *R*_1_ = the radius of the growth zone (cm) of the pathogen in contact with the microalgae suspensions.

### 3.4. In Vivo Evaluation of the Biocontrol Potential of Microalgae

The in vivo trials were conducted taking into consideration the results obtained in vitro: only the algal concentrations in the in vitro trials with equal and significantly higher inhibition values than the positive controls (*p* < 0.05) were validated by the in vivo trial. The experiment took place in an unheated polyethylene greenhouse from March to May under natural photoperiod conditions. The humidity and air temperature were increased by growing the plants under a tunnel inside the greenhouse. The plants were irrigated by micro-sprinklers to maintain high humidity and moisture in the substrates. The mean air temperature was ≥20 °C, and the average humidity was above 57%. This in vivo method was an adaptation from Biondi et al. [60]. Firstly, the peat pH (Hansa Torf Floragard, Germany) was corrected to near 6.5 by adding 4.5 g L^−1^ of fine calcium carbonate (CaCO_3_).

Part of the substrate was sterilized in a dry chamber (MMM Medcenter Venticell, Germany) at 60 °C for one week to eliminate most microorganisms (thermally treated peat).

To inoculate the substrate with *F. oxysporum* f. sp. *spinaciae*, the dilution plate technique was used, where a suspension was prepared from a pure culture by adding 50 mL of sterile deionized water and using a cell spreader to mix the fungi in the water. This mixture was diluted ten times, and 200 µL from this mixture was taken and put in the hemocytometer. After that, the spores were counted, and the colony-forming units (CFU) per millilitre were calculated. A suspension with a final concentration of 7.5 × 10^5^ CFU mL^−1^ was prepared to ensure sufficient spores to inoculate 3 L of peat [61]. The bag with the sterilized peat was from the dry chamber (MMM Medcenter Venticell, Germany) and cooled down to room temperature. An amount of 3 L of the peat was put in a sterile bucket, and 204 mL of the suspension with the disease was added. The same steps were carried out for the non-heat-treated peat. Afterwards, the inoculated substrate bags (with and without thermic treatment) were put in the dry chamber at 25 °C for one week to stimulate the growth of *F. oxysporum* f. sp. *spinaciae*. At the same time, spinach seeds (*Spinacia oleracea* L. ‘Lizard’) were sown in vermiculite.

After a week, 100 mL pots were filled with the two substrates: unsterilized and sterilized commercial peat, both containing the phytopathogen. The experimental design consisted of 6 treatments per microalgae and substrate: 3 concentrations, 2 positive controls (synthetic fungicide and a commercial biopesticide), and 1 negative control (distilled water), distributed in four randomized complete blocks. Each treatment consisted of 4 pots, for a total of 96 pots.

To evaluate the biocontrol potential of *C. vulgaris* and *T. obliquus*, aqueous suspensions were prepared at three different concentrations, as determined in previous trials (Table 4).

Weekly, 20 mL of the algal suspensions was applied directly to the substrate. The positive controls were applied to the substrate according to the manufacturer’s guidelines: every other week, Rovral at 1.0 g L^−1,^ and T34 at 0.01 g L^−1^. Water was used as the negative control, and the same volume was applied weekly. The three algal concentrations and the three controls were each applied four times.

### 3.5. Suppressive Evaluation Assessments

Weekly, the visual effects of the disease severity were evaluated according to a scale from 1 to 4, adapted from Baayen and Van der Plas [62], where 1 = no injuries to the plant, 2 = minor injuries, 3 = major injuries, and 4 = post-emerging damping off (Figure 6).

The incidence score was valued at 0 for plants without symptoms and 1 for plants with symptoms. The trial ended when all the plants with a certain treatment died. At the end of the trial, the area under the disease progress curve (AUDPC) was determined [63], according to the following equation:$$\sum_{i=1}^{n}=(\frac{y_{i}+y_{i+1}}{2})(t_{i+1}-t_{i})$$
where the components are as follows:*y**_i_* = Severity of the disease (when the first symptom is observed);*y*_*i*+1_ = Severity of the disease (at the end of the trial);*t**_i_* = Time (the beginning of the trial in days);*t*_*i*+1_ = Time (the end of the trial in days).

### 3.6. Chemical Characterization of Microalgae

#### Total Polyphenol Content

The total amount of polyphenols present in the microalgae biomass was determined according to the Folin–Ciocalteu assay adapted from Velioglu et al. [64]. Briefly, extract samples were prepared as follows: 150 mg of dry weight dry alga biomass was resuspended in 1 mL of methanol in tubes; 0.7 g of glass beads (425–600 µm) was added to the tubes; cell lysis was performed by placing the tubes in a Retsch MM400 homogenizer and bead milled at a frequency of 30 Hz for 5 min. The samples were then centrifuged at 8160× *g* (Scansci Hermle Z167M, Portugal) for 5 min. The supernatant was collected and transferred to a previously weighed glass tube, and the pellet was repeatedly resuspended until the supernatant became transparent. Afterwards, the entire supernatant was filtered with a 0.2 μm nylon filter and dried using nitrogen flow (stuart SBHCONC/1). The dried extract was weighed, the yield was calculated and resuspended in DMSO to achieve a final concentration of 20 mg mL^−1^, and the extracts were stored at −20 °C for no longer than 4 weeks until analysis.

The assay was conducted using 96-well plates. An extract aliquot of 5 μL was added to the wells, mixed with 100 μL of the Folin–Ciocalteu reagent, and incubated at room temperature for 10 min. The reaction was then completed by adding 100 μL of sodium carbonate (75 g L^−1^), and the plate was incubated in the dark for 90 min at room temperature. Finally, the absorbance was read at 725 nm in a microplate reader (BioTek Synergy 4, Shoreline, WA, USA). The results were expressed as gallic acid equivalents (GAE) in milligrams per gram of extract in dry weight (DW) using a calibration curve of gallic acid at concentrations from 0.001 to 2 mg mL^−1^ (*R*^2^ = 0.995, *p* < 0.01).

### 3.7. Statistical Analysis

Further insight into the results was obtained with the Statistical Package for the Social Sciences (SPSS v. 29). Data gathered from the in vitro trials were evaluated by a regression analysis. The comparison between the treatments in the in vivo trial was performed by a Multi-factor Analysis of Variance (ANOVA) followed by the Tukey–Kramer test.

## 4. Conclusions

This study showed that suspensions of the microalgae *Chlorella vulgaris* and *Tetradesmus obliquus* suppressed the growth and development of *F. oxysporum* f. sp. *spinaciae* in vitro. Despite affecting its development, this trial has shown that there is a dose effect that still needs more tests to be understood. Moreover, the in vivo trials showed that both microalgae were able to control the disease in spinach. The microalgae polyphenolic content can explain the ability to control the phytopathogenic fungi; however, it is unlikely to be the only compound responsible for this effect. As *F. oxysporum* is one of the plant pathogenic fungi most difficult to control, the findings of this study show that aqueous suspensions of algae biomass applied through irrigation can be an eco-friendly alternative to the use of agrochemicals, achieving more sustainable agriculture. More studies need to be conducted to better understand the fungi–alga relationship and how this interaction works.

## Figures and Tables

**Figure 1 plants-13-01697-f001:**
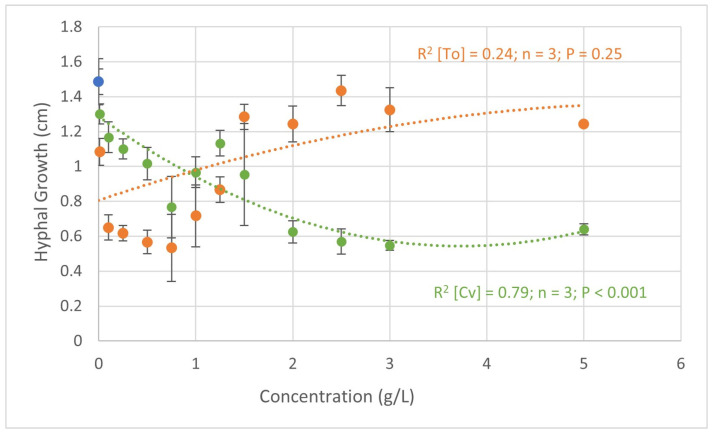
Quadratic regression analysis of the relationship between the hyphal growth (cm) of *Fusarium oxysporum* and the concentrations of *C. vulgaris* (green) and *T. obliquus* (orange). R^2^—coefficient of determination; P—probability; and n—number of observations. In blue is represented the hyphal growth in the negative control.

**Figure 2 plants-13-01697-f002:**
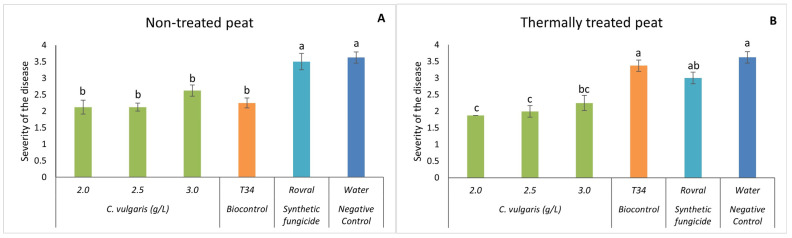
In vivo severity of *F. oxysporum*-induced disease in spinach plants. (**A**) Disease severity results in non-treated peat with *C. vulgaris*. (**B**) Disease severity results in thermally treated peat with *C. vulgaris*. The in vivo results of disease severity were recorded at the end of the trial. For each treatment, values are mean ± SE (*n* = 8). Bars with different letters indicate significantly different values at *p* < 0.05 by the Tukey–Kramer test.

**Figure 3 plants-13-01697-f003:**
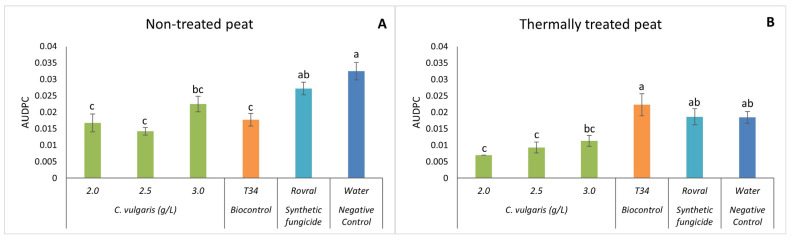
AUDPC results of *F. oxysporum*-induced disease in spinach treated with *C. vulgaris*. (**A**) AUDPC results in non-treated peat treated with *C. vulgaris*. (**B**) AUDPC results in thermally treated peat treated with *C. vulgaris*. AUDPC was calculated for each treatment; values are mean ± SE (*n* = 8). Bars with different letters correspond to significantly different values at *p* < 0.05 using the Tukey–Kramer test.

**Figure 4 plants-13-01697-f004:**
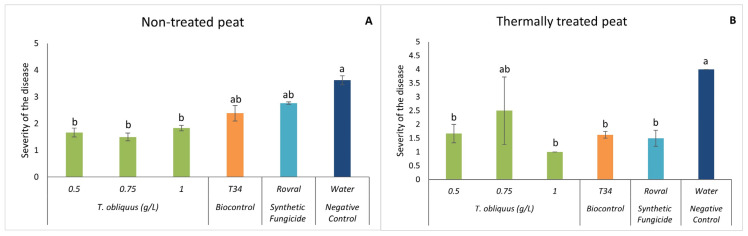
In vivo severity of *F. oxysporum*-induced disease in spinach plants. (**A**) Severity results in non-treated peat with *T. obliquus*. (**B**) Severity results in thermally treated peat with *T. obliquus*. In vivo results of severity were recorded at the end of the trial. For each treatment, values are mean ± SE (*n* = 8). Bars with different letters correspond to significantly different values at *p* < 0.05 using the Tukey–Kramer test.

**Figure 5 plants-13-01697-f005:**
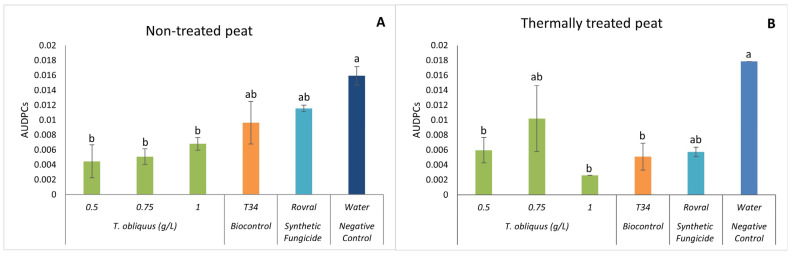
AUDPC results of *F. oxysporum*-induced disease in spinach treated with *T. obliquus*. (**A**) AUDPC results in non-treated peat treated with *T. obliquus*. (**B**) AUDPC results in thermally treated peat treated with *T. obliquus*. AUDPC was calculated for each treatment, values are mean ± SE (*n* = 8). Bars with different letters correspond to significantly different values at *p* < 0.05 using the Tukey–Kramer test.

**Figure 6 plants-13-01697-f006:**
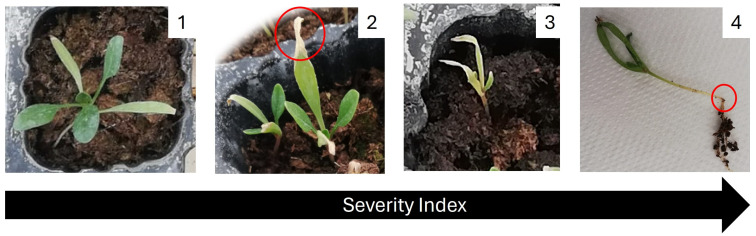
Visual scale of disease severity in spinach. Severity index was defined by the following: 1—healthy plant; 2—minor injuries; 3—major injuries; and 4—post-emerging damping off. In red, the symptoms of the disease caused by the *F. oxysporum* f. sp. *spinaciae*.

**Table 1 plants-13-01697-t001:** Populations of microorganisms in the substrates.

	CFU g^−1^ Substrate					
	Thermally Treated Peat			Non-Treated Peat		
	Fungi	Actinomycetes	Bacteria	Fungi	Actinomycetes	Bacteria
Peat	0	0	0	2.81 × 10^7^	2.60 × 10^7^	5.92 × 10^7^

CFU, colony-forming unit.

**Table 2 plants-13-01697-t002:** Incidence of *F. oxysporum* in spinach, 35 days after inoculation, in both substrates.

	Treatments	Concentrations (g L^−1^)	Non-Treated Peat	Thermally Treated Peat
Disease Incidence (%)	*C. vulgaris*	2.0	88 ^a^	88 ^a^
		2.5	100 ^a^	88 ^a^
		3.0	100 ^a^	100 ^a^
	Rovral		100 ^a^	100 ^a^
	T34		100 ^a^	100 ^a^
	Water		100 ^a^	100 ^a^

For each treatment, values are mean (n = 8). For each peat type, treatments with different letters indicate significantly different values at *p* < 0.05 by the Tukey–Kramer test. T34, biocontrol agent; Rovral, commercial synthetic fungicide.

**Table 3 plants-13-01697-t003:** Incidence of *F. oxysporum* in spinach 47 days after inoculation in both substrates.

	Treatments	Concentrations (g L^−1^)	Non-Treated Peat	Thermally Treated Peat
Disease Incidence (%)	*T. obliquus*	0.5	42 ^c^	67 ^a^
		0.75	50 ^ab^	50 ^a^
		1.0	50 ^ab^	0 ^a^
	Rovral		94 ^ab^	50 ^a^
	T34		60 ^ab^	67 ^a^
	Water		100 ^a^	100 ^a^

For each treatment, values are mean (n = 8). For each peat type, treatments with different letters indicate significantly different values at *p* < 0.05 by the Tukey–Kramer test. T34, biocontrol agent; Rovral, commercial synthetic fungicide.

**Table 4 plants-13-01697-t004:** Microalgae concentrations in the treatments used for the in vivo trials.

	Treatments (g L^−1^)					
Microalgae	Non-Treated Peat			Thermally Treated Peat		
*C. vulgaris*	2.0	2.5	3.0	2.0	2.5	3.0
*T. obliquus*	0.5	0.75	1.0	0.5	0.75	1.0

## Data Availability

The data presented in this study are available on request from the corresponding author.

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
