# Peer review of "Chlorella vulgaris and Tetradesmus obliquus Protect Spinach (Spinacia oleracea L.) against Fusarium oxysporum"

_plants, 2024, doi:10.3390/plants13121697_

Round 1

Reviewer 1 Report

Comments and Suggestions for Authors

Reviewer’s comment

Chlorella vulgaris and Tetradesmus obliquus Protect Spinach against Fusarium oxysporum by Viana et al., is an interesting study that demonstrates the effectiveness of utilizing Chlorella vulgaris and Tetradesmus obliquus as biocontrol agents. Thus the study evaluates Chlorella vulgaris and Tetradesmus obliquus as biocontrol agents against the phytopathogenic fungus Fusarium oxysporum. Their in Vitro results demonstrates that both microalgae showed inhibitory effects on the fungus, with C. vulgaris achieving 40-50% growth inhibition and T. obliquus achieving 42-64%. In their in vivo validation of the study, employing spinach, they showed that there was a reduction in disease severity and incidence when treated with microalgae suspensions. This work supports the potential use of microalgae as sustainable biocontrol agents in agriculture, aligning with initiatives to reduce chemical pesticide use. The methods used are adequate, the discussion is sufficient and succinct.

Minor Revisions

The figures require revision to enhance quality. Each figure should have 300 pi at least.

Figure 1A is poorly labeled, see Figure 1B (the letter B is inserted inside the Figure box).

Figure 1 X-axis labels are faintly labeled and hard to read. Revise appropriately. Revise similarly throughout in all other figures.

The first sentence of the conclusions requires revision for better comprehension. “This study showed that both microalgae, Chlorella vulgaris and Tetradesmus obliquus, the growth and development of F. oxysporum and were able to control the symptoms of its disease in spinach.”

Line 90. “(Error! Reference 90 source not found.)” Revise and put in the requisite reference(s). Please revise throughout the manuscript.

Line 102. “C. vulgaris, T. obliquus” Italicize these scientific names and revise throughout the body of the work.

Author Response

Dear Reviewer,

Thank you very much for taking the time to review the manuscript “Chlorella vulgaris and Tetradesmus obliquus Protect Spinach (Spinacia oleracea L.) against Fusarium oxysporum”. Please find the detailed responses below and the document attached corresponding revisions/corrections highlighted/in track changes in the re-submitted files.

Reviewer 2 Report

Comments and Suggestions for Authors

Dear Authors,

In this article, the authors use two microalgae and assess their potential to prevent infection by a fungus in spinach.

Majors:

-To my understanding, the introduction falls short; it lacks the necessary depth. For instance, there are relevant studies on microalgae producing a variety of metabolites that have been shown to possess antifungal properties, none of which are cited. I'll give just one example; you don't need to cite it: doi.org/10.1007/s10811-023-03015-x.

-The Fig 1. has numerous editing errors. Letter A is outside the frame, there is no Y-axis, and the concentrations of the positive controls are not indicated.

-I  can't find the basis on which those specific two positive controls were chosen Rovral and T34?

-This error is repeated throughout the entire paper; the authors have not taken due care to submit the final version, which does not give a good impression.L90: (Error! Reference  source not found.). L187: Error! Reference source not found.

-L113: “Similar findings were found in a study [9] on the potential of different extracts of C.  vulgaris. The extracts used different solvents, namely acetone, diethyl ether, and methanol.”  This is an example of insufficient explanation regarding the papers cited. I don't understand what the authors mean.  What importance do solvents have? What solvent have you used in your study? It should be indicated.

-L128-L133. This is a big mistake: the authors completely overlook commenting that there is a significant and clear negative effect of T. obliquus at concentrations above 1.25, Fig 1, they don't even provide an explanation.

-I don't understand the results of Table 2. How is it possible that water has the same incidence in the disease as the positive control? And,  It doesn't show any type of statistical error as the table 1. And what's even worse, Table 2 is not cited anywhere in the text!!!!

-The same with table 3. I don't understand how it's possible for them to present data without commenting on it?

-The statistical errors in Figure 4 and 5 are so significant that it lacks any scientific value

-L392: Total Polyphenols Content?? But I don't see any results presented regarding the polyphenol content?

Minors:

L84: in in vivo assays

Author Response

(The authors gave the same response as above.)

Reviewer 3 Report

Comments and Suggestions for Authors

The manuscript entitled „Chlorella vulgaris and Tetradesmus obliquus Protect Spinach (Spinacia oleracea L.) against Fusarium oxysporum”, submitted for evaluation to Plant, presents the results concerning the effect of Chlorella vulgaris and Tetradesmus obliquus as new ecological agents protecting spinach crops against fungal (F. oxysporum) infection.

 In general, the quality of submitted communication is acceptable. English is clear and understood, methodology in general is repeatable and described in clear way. Results are also clearly explained.

I have two minor comment, listed below:

 COMMENTS TO Authors

1.      Please provide the lacking citations.

2.      Please note that names of microorganisms should be written in italics (e.g. legend to Table 2)

Author Response

Dear Reviewer,

Thank you very much for taking the time to review the manuscript “Chlorella vulgaris and Tetradesmus obliquus Protect Spinach (Spinacia oleracea L.) against Fusarium oxysporum”. Please find the detailed responses below in the document attached and the corresponding revisions/corrections highlighted/in track changes in the re-submitted files.

Reviewer 4 Report

Comments and Suggestions for Authors

The sustainable management of Fusarium disease in spinach through algal bioagent is a good attempt and of great interest to the researchers. However, this manuscript should be carefully revised from statistical perspective. There is overlapping error bars clearly depicting in the bar graphs but still it has been shown significantly different. There are major mistakes in presenting the data and statistical analysis. Also, there are certain missing elements in this work which should be added in the respective section.

1.      The trail details are missing, plot size, date of planting, inoculation, data recording replicates all these should be added in material method section.

2.      Likewise, the details of fungal pathogen and bioagents should be elaborated by providing the id of genomic accession or culture collection id.

3.      The major limitation is lack of any figure of disease mitigation under in vitro and in field condition which would add value to this work

4.      For how many years the experiment was performed or it is a single year study?

5.      Please incorporate recent studies

6.      Correct the scientific names across the manuscript.

7.      What about yield related data in control and treated plots.

8.      Please re-analyse the data and do appropriate interpretation of actual differences in respective treatments. 

Comments on the Quality of English Language

Minor editing is required.

Author Response

(The authors gave the same response as above.)

Round 2

Reviewer 2 Report

Comments and Suggestions for Authors

I believe that the authors have adequately addressed all my comments and suggestions, and I accept the paper in its current version

Reviewer 4 Report

Comments and Suggestions for Authors

The changes made as per comment are satisfactory.